# Assessment of Bone Height Changes Based on the Cone–Beam Computed Tomography Following Intentional Replantation for Periodontally Compromised Teeth

**DOI:** 10.3390/medicina59010040

**Published:** 2022-12-25

**Authors:** So-hyun Park, Seung-Heon Paek, Bongju Kim, Jung-Tae Lee

**Affiliations:** 1Department of Advanced General Dentistry, Dankook University Jukjeon Dental Hospital, Yongin-si 16890, Republic of Korea; 2Dental Life Science Research Institute, Seoul National University Dental Hospital, Seoul 03722, Republic of Korea; 3Department of Periodontics, One-Stop Specialty Center, Seoul National University, Dental Hospital, Seoul 05698, Republic of Korea

**Keywords:** intentional replantation, diagnosis, periodontal involvement, periodontitis

## Abstract

*Background and Objectives*; This study aimed to evaluate the clinical outcomes and bone changes before and after intentional replantation (IR) for periodontally compromised teeth by using cone–beam computed tomography (CBCT). *Materials and Methods*; Fourteen periodontally involved teeth were selected for IR. A preoperative orthodontic procedure was performed to apply luxation, and the tooth was then gently extracted. Retrograde filling of the root-end canal was performed. Once the tooth was repositioned in the socket, it was splinted with the adjacent tooth. After three months, prosthetic restoration was performed. *Results*; Clinical parameters and CBCT images were obtained before and after the IR procedure. The height of the alveolar bone was measured on the CBCT images by using software. Most preoperative symptoms, including pain, mobility, probing depth (PD) and bleeding on probing (BOP), significantly decreased after IR (pain: 4.71 to 1.00; mobility: 1.36 to 0.29; PD: 5.60 to 2.85; BOP: 3.50 to 0.79). CBCT analysis indicated an increase in bone height after IR (the amount of change: maxilla, 4.00; mandible, 1.95). *Conclusions*; A previous study reported that IR for periodontally involved teeth is quite limited. However, in this study, IR of periodontally compromised teeth showed favorable results in clinical and radiographic evaluations, suggesting that IR may be an alternative to extraction of teeth with periodontal disease.

## 1. Introduction

The intentional replantation (IR) technique consists of tooth extraction with an intentional atraumatic method, extra-alveolar evaluation of the root surface, root canal treatment, and reinsertion of the tooth into its original position [1,2]. The advantage of IR is that it can be used to directly evaluate and treat the tooth outside the alveolar socket. Despite advancements in surgical root canal treatment, IR has been used in some challenging cases involving endodontic–periodontal lesions and lesions with difficult accessibility (maxillary sinus, perforation of root, and palatal groove) [3]. IR also has the advantage of being cost effective compared to dental implants [4]. Therefore, IR can be considered another choice for preserving teeth and may be an alternative to tooth extraction and implant placement [5].

Previous studies of IR reported success rates ranging from 50% to 95% [6,7]. Wu and Chen demonstrated that survival rates were decreased in the first year, but these values were stable at 4 years with 82.8% survival rate [8]. Recent systematic studies have reported survival rates of approximately 90% for IR [4]. Surgical variables including orthodontic techniques, root-end bio-materials, tissue-regeneration agents, and anti-inflammatory medication have resulted in an increase in the survival rate associated with IR. Choi et al. suggested that a preoperative orthodontic treatment helped increase the IR success rate with inducing atraumatic tooth extraction and periodontal ligament (PDL) healing [9]. Wang et al. demonstrated the extraoral time period seems to be a critical factor to the success of the IR procedure [10]. Over 30 min of extra-alveolar time result in nonviable PDL cells due to drying. Less than fifteen minutes of extraoral time are recommended to clinicians in previous studies [11]. Mineral trioxide aggregate (MTA) is a suitable retro-filling material which has properties of biocompatibility and calcific barrier formation by reinforcing osteogenic PDL cell differentiation [12]. The 4-methacryloxyethyl trimellitate anhydride/methyl methacrylate-tri-n-butylborane (4-META/MMA-TBB) dentin-bonded resin is also recommended as a proper sealing material to repair damaged roots of teeth [13].

IR for teeth with periodontal disease is a challenging procedure for clinicians. A previous study documented that periodontally involved teeth with over 6 mm on the two sites may be negative on tooth survival [3]. Although IR is considered to be contraindicated in teeth with moderate-to-severe periodontal disease [14], one previous study reported successful results in periodontally compromised teeth [3]. Manikar et al. reported that one of the indications for IR is periodontally compromised teeth [4], and previous studies have shown favorable results for IR in teeth with periodontally poor prognosis [15,16]. In addition, guided tissue regeneration (GTR) was a proposed good choice for teeth with inadequate bony support, and it maintains space for osteoblastic cells by preventing the migration of epithelial cell into PDL space [17].

Radiographic images can provide anatomical information regarding aspects such as the marginal bone level, root distortion, and the presence of impacted teeth [18]. Panoramic and intraoral periapical radiography is the most commonly used radiographic method for such assessments [19]. However, these conventional radiographic measurements with two-dimensional plane show limitations that can lead to misinterpretation, such as image magnification and distortion by projection errors [20]. Cone–beam computed tomography (CBCT) has been proposed to overcome these limitations. Lee et al. evaluated the marginal bone level after GTR and open flap procedures using CBCT [21]. Previous studies have suggested that CBCT is helpful in evaluating the anatomical structure of teeth and surrounding areas before and after IR procedures [22,23].

One review article suggested that the literature on IR for periodontally involved teeth is quite limited [24]. Therefore, the present study aimed to evaluate the clinical outcomes and bone changes before and after IR in periodontally compromised teeth by using CBCT. The null hypothesis of this study was that the bone height showed no difference before and after IR for periodontally compromised teeth.

## 2. Materials and Methods

### 2.1. Patients

The study was conducted in accordance with the guidelines of the Declaration of Helsinki and approved by the Institutional Review Board for Clinical Research at the Dankook University Jukjeon Dental Hospital (approval No. 201612-001-002). From April 2015 to August 2016, 14 patients (eight males and six females; mean age, 46.79 years) who required tooth retention, but not extraction, visited the Jukjeon Dental Hospital of Dankook University. Fourteen teeth of these patients were selected for IR, which included extraction of the periodontally involved tooth (Table 1).

Patients who met the following criteria were included in the present study [25,26]:No severe systemic disease (hypertension, osteoporosis, diabetes mellitus, etc.);Fifty percent or more radiographic bone loss, 5 mm or deeper probing depth (PD) and grade Ⅲ mobility due to severe periodontal destruction (Miller’s classification: measured as M0–M3, where M0 is physiologic mobility, M1 is slightly increased mobility, M2 is considerably increased mobility without impairment of function, and M3 is extreme mobility) [27];The mobility of the adjacent teeth was less than grade II and the distance between teeth was less than 2 mm;Patient preference for tooth retention rather than extraction.

### 2.2. Periodontal Examination

A single dentist (J.T.L) performed periodontal examinations; PD at six sites/tooth using a University of North Carolina (UNC)-15 periodontal probe, bleeding on probing (BOP) (defined as bleeding for 20 s after probing), and mobility [25]. Pain value was measured by a visual analog scale (VAS) obtained from questions for patients [28].

### 2.3. Radiographic Evaluation

A 3D CBCT scanner (Kodak 9500, Carestream Health, Rochester, NY, USA), which provided a grayscale 14-bit image with a voxel size of 0.2 mm per side, was used in this study. CBCT images were viewed using 3D imaging software (OnDemand 3D, Cybermed Co., Seoul, Republic of Korea) [29].

### 2.4. Measurement of Bone Height

Radiographic images were evaluated by two dentists (S.-h.P.and S.-H.P.). To ensure data reliability, two observers calibrated the bone height and reviewed selected dental CBCT images and periapical radiographs on the basis of a previous study involving hopeless teeth [30]. A serial number was assigned to each section of the CBCT images to prove the consistency of the measurements. [21] All pre- and postoperative bone height measurements were performed by the first tester (S.-h.P.). The second tester (S.-H.P.) also evaluated with the same serial number on the same image. The study measurements were recorded as follows: maxillary tooth bone height was defined as the distance from the top of the bone level to the floor of the sinus, and mandibular tooth bone height was defined as the distance from the top of the bone level to the roof of the inferior alveolar canal [31]. The standard bone level was considered to be between the top of the alveolar bone level and the cementoenamel junction (CEJ). The distance between the CEJ and the floor of the sinus or between the CEJ and the roof of the inferior alveolar canal was measured before the IR procedure. After the IR procedure, bone height was measured by determining the distance between the CEJ and the floor of the sinus or between the CEJ and the roof of the inferior alveolar canal. The bone height was adjusted by comparing the preoperative and postoperative lengths (Figure 1).

Bone height change in the maxilla (mm)
= (h′s − h′bs) − (hs − hbs);

(Maxilla: hs = CEJ to the floor of the sinus; hbs = top of the alveolar bone level to the floor of the sinus; h= preoperative length; h’= postoperative length).

Bone height change of the mandible (mm)
= (h′c − h′bi) − (hc − hbi);

(Mandible: hc = CEJ to the inferior alveolar canal; hbi = top of the alveolar bone level to the roof of the inferior alveolar canal; h= preoperative length; h’= postoperative length).

### 2.5. Pre-Operation Orthodontic Procedure

For atraumatic extraction, an orthodontic button was attached to the buccal surface of the tooth requiring IR in two patients [32], and a nickel–titanium wire was ligated to apply 50 g of orthodontic extrusive force to the tooth for 2–3 weeks (Figure 2). In four patients, an orthodontic separator was inserted between both sides of the surface of the tooth requiring IR (Table 2) [5].

### 2.6. Surgical Procedure

Non-surgical periodontal therapies (scaling and root planing) were administered before IR. All IR treatments were completed by the same dentist who performed the periodontal examinations (J.T.L). In most cases, apical bone resorption had occurred because of endodontic–periodontal lesions. During each procedure, the affected tooth was extracted as gently as possible and placed on a sterile, moist gauze. The socket was rinsed with sterile saline solution and any granulation tissue at the bottom of the socket was removed if needed [33]. The total extra-alveolar manipulation time was maintained within 15 min [2,6]. Any remaining granulation tissue, calculus, affected PDL, and necrotic cementum on the tooth root surfaces were gently removed using a hand scaler and Gracey curettes (Figure 3A–D).

### 2.7. Retrofilling Procedure and Splinting Teeth

A 3-mm root-end preparation was achieved using an ultrasonic tip. Retrograde filling of the root-end canal was performed with MTA or 4-META/MMA-TBB dentin-bonded resin [24]. After root resection, a space between the alveolar bone floor and apical root tip was formed using guided bone regeneration (GBR) with demineralized freeze-dried bone (DFDBA; Human Cortical Powder, Demineralized, DIZG, Berlin, Germany). In nine patients, after the extracted tooth was repositioned in the socket, the bone graft was completed. Once each tooth was repositioned into the socket, it was splinted with the adjacent teeth for 12 weeks to support periodontal healing (Table 2) [34]. Occlusal adjustment was performed to avoid tight occlusion [35]. Post-operative medication was prescribed 3 times daily for 1 week as follows: amoxicillin and clavulanic acid 375 mg (Augmentin, Ilsung Pharma Co., Seoul, Republic of Korea), naproxen sodium (Anaprox, Jongeun Dang Pharmceutical, Co., Seoul, Republic of Korea), and almagate (Almagel, Yuhan Pharma Co., Seoul, Republic of Korea). Oral rinsing with 0.12% chlorhexidine gluconate (Hexamedin, BukwangPharmaceutical, Ansan, Republic of Korea) was instructed for 30 s twice a day. After three months, prosthetic restoration was performed (Figure 3E–J).

### 2.8. Statistical Analysis

Statistical analysis was performed using SPSS software (SPSS version 18.0, Chicago, IL, USA). The Mann–Whitney U test was used to compare alveolar bone height change amount differences between maxilla and mandible. Wilcoxon signed-rank test was performed to evaluate clinical parameters and bone height before and after IR procedure. P or corrected *p* (Pc) values < 0.05 were considered significant.

## 3. Results

### 3.1. Clinical Observation

Fourteen teeth in 14 patients were treated, including one maxillary central incisor, four maxillary molars, and nine mandibular molars. All patients were satisfied with the outcomes of the IR treatment and tolerated the IR procedures without any complications. The radiopacity of the apical region increased after IR treatment (Figure 3K,L). Clinically, the color and texture of the gingival tissue had changed to pink and firm, respectively. The mean preoperative pain VAS score was 4.71 (SD: 2.84), and the mean postoperative pain VAS score was 1.00 (SD: 1.18), showing a mean reduction of 3.71 (*p* < 0.05). Miller’s classification of tooth mobility [19] showed a preoperative mean mobility of 1.36 (SD: 1.34) and a postoperative mean mobility of 0.29 (SD: 0.47), with a mean reduction of 1.07 (*p* < 0.05). The PD value was calculated as the mean value obtained by probing at six sites. The mean preoperative PD was 5.60 (SD: 2.02) and the mean postoperative PD was 2.85 (SD: 1.21) with a mean reduction of 2.75 (*p* < 0.05). The BOP value calculated the mean value on six sites showing bleeding. The mean preoperative BOP was 3.50 (SD: 1.56), and the mean postoperative BOP was 0.79 (SD: 0.70), showing a mean reduction of approximately 2.71 (*p* < 0.05) (Table 3).

### 3.2. Changes in Bone Height before and after IR

Bone height was measured by determining the mean values for six sites on maxillary and mandibular teeth. The mean preoperative and postoperative bone heights of the maxillary teeth were 43.29 mm (SD: 28.45) and 47.29 mm (SD: 33.04), respectively. The mean preoperative and postoperative bone heights of the mandibular teeth were 115.50 mm (SD: 42.98) and 113.55 mm (SD: 39.45), respectively. The width of the bone change was 4.00 mm in the maxilla and 1.95 mm in the mandible. No significant differences were observed in bone height (*p* > 0.05) (Table 4 and Figure 4).

## 4. Discussion

The present study showed that the clinical outcomes and bone changes determined using CBCT before and after IR for periodontally compromised teeth were different. Our findings indicated that the VAS scores for pain, tooth mobility, PD, and BOP decreased after the IR procedure (mean values: pain, 4.71 to 1.00; tooth mobility, 1.36 to 0.29; PD, 5.60 to 2.85; and BOP, 3.50 to 0.79). These parameters showed significant differences before and after IR. Bone heights of the maxilla and mandible increased (maxilla: 4.00 mm; mandible: 1.95 mm) after treatment. This finding may be explained by the findings of a previous study, in which Zhang et al. reported that PD and alveolar bone height decreased after IR of periodontally involved hopeless teeth [26]. However, the maxillary and mandibular bone heights did not differ before and after IR in this study.

IR involves intentional extraction and reinsertion of a tooth into the socket after endodontic manipulation and/or obturation of the canals [36]. By performing this procedure extraorally, the root surface can be visualized and treated easily without damaging the adjacent periodontal complex. In addition, all local factors in teeth that cause periodontal inflammation, attachment loss, and alveolar bone resorption can be eliminated. IR can yield a better prognosis in patients with endodontic–periodontal lesions. Using the IR procedure, the origin of periodontal problems (such as subgingival calculus and salivary parameters, etc.) as well as endodontic problems can be easily removed [11,37]. Periodontal involvement is a major contraindication for IR. Grossman [1] suggested that IR was contraindicated in teeth with extensive mobility or alveolar bone destruction, or when the septal bone (at the bifurcation) was destroyed or missing in cases involving the posterior teeth. Bender and Rossman [35] reported that IR is not recommended for teeth with excess mobility, furcation involvement, or gingival inflammation. However, some studies have shown favorable results for IR in teeth with periodontal involvement. Lu [38] reported successful clinical results with IR in periodontally involved teeth after intentionally replanting an endodontically mistreated and periodontally involved mandibular first molar. The tooth was maintained under asymptomatic and functional conditions for 32 months. Demiralp et al. [30] performed IR for 15 periodontally involved hopeless teeth and followed the outcomes for 6 months. Thus, IR has been suggested as an alternative approach in cases showing advanced periodontal destruction with extraction as the only other treatment option.

In this study, most preoperative symptoms, including pain, mobility, PD, and BOP, decreased significantly, and the patients showed no occurrence of generalized root resorption, no evidence of periapical rarefaction, and no presence of slight localized evidence of root resorption. According to one previous study, the presence of periapical lesions is correlated with the maxillary sinus mucosa thickness (MSMT) [39]. In Figure 4A,B,E,F, the MSMT decreased after the IR procedure. Although the teeth were functioning normally, the goal of the treatment is to achieve an asymptomatic status with healthy gingiva for ≥5 years, a significant reduction in pocket depth, and new bone formation for long-term success. Failure of IR presents with evidence of resorption, rarefaction, or discomfort. The key factor for the success of IR is a viable PDL [40]. In cases showing severe periodontal involvement, maintenance of the viability of PDL is challenging, and the healing process after intentional extraction and reinsertion may not be encouraging. Without a viable PDL, ankylosis is a common complication of replanted teeth that leads to gradual resorption of hard tissues and their replacement by bone. The teeth evaluated in this study met the criteria of success listed above, and showed good functioning with no patient discomfort. The patients showed no gingival recession, pathological periodontal pocket formation, evidence of additional marginal bone loss, or evidence of ankylosis and root resorption. Bone grafting was performed with IR in this study. Bone grafting seems to maintain space for osteoblasts by precluding contact between the connective tissue and PDL [41]. Zufia et al. reported that IR, including bone grafting, should be considered when the bone support is insufficient [17].

This study differs from previous studies in that it compared changes before and after IR by using CBCT. Although conventional clinical measurements such as PD, BOP, and mobility are good evaluation factors, they have limited reliability [42]. Intraoral periapical and panoramic radiographs cannot easily yield accurate images because of the 2D plan images. CBCT has been used to overcome the limitations of conventional radiographic methods. With a multidimensional view, CBCT can reveal the anatomy of the tooth and its surrounding structures [43,44]. Previous studies have suggested that CBCT may be helpful for IR procedures [27,30,31]. Granichi et al. strongly recommended the use of CBCT for diagnosis and treatment in cases requiring IR [45]. The radiation in CBCT has been demonstrated to be 15 times lower than that in conventional radiography [46]. This low radiation dose may be another advantage of CBCT in such examinations.

Nevertheless, the sample size used in this study was insufficient, and a larger sample size is needed to evaluate IR of teeth affected by periodontal diseases. Future studies should aim to include long-term follow-up assessments and evaluation of various factors (anterior or posterior position) affecting IR. In addition, research on the varying degrees of periodontal destruction is required to obtain more clarity about guidelines for applying IR to more “last resort” cases.

## 5. Conclusions

Within these limitations, this study suggests the following:IR of periodontally compromised teeth is an alternative treatment for extraction;In particular, IR seems to be proposed as a method to treat endodontic-periodontal lesions;GTR is recommended as a proper choice for teeth with inadequate bony support.

## Figures and Tables

**Figure 1 medicina-59-00040-f001:**
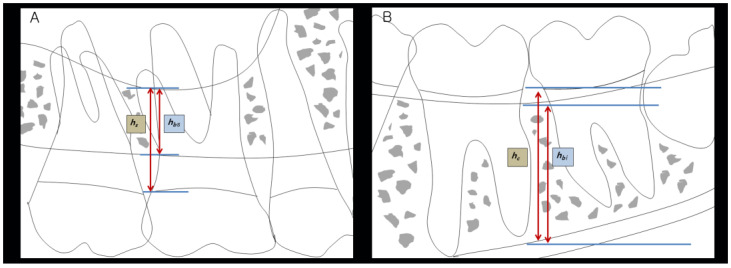
Measurement of bone height. (**A**) Maxilla: (h′s − h′bs) − (hs − hbs) (hs = CEJ to floor of the sinus; hbs = top of the alveolar bone level to floor of the sinus); (**B**) Mandible: (h′c − h′bi) − (hc − hbi) (hc = CEJ to inferior alveolar canal; hbi = top of the alveolar bone level to the roof of the inferior alveolar canal; h= preoperative length; h’= postoperative length). CEJ: cementoenamel junction.

**Figure 2 medicina-59-00040-f002:**
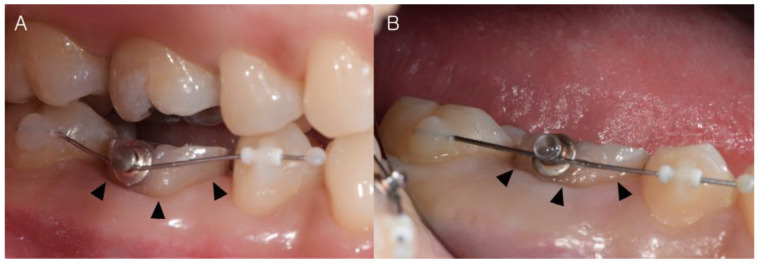
Pre-operative orthodontic procedure. (**A**) An orthodontic button was attached toward the buccal surface of the tooth requiring IR for atraumatic extraction. A nickel–titanium wire was ligated for applying 50 g of orthodontic extrusive force onto the tooth. (**B**) After 2–3 weeks, the tooth was extruded in the coronal direction (Arrow: movement of teeth in the coronal direction).

**Figure 3 medicina-59-00040-f003:**
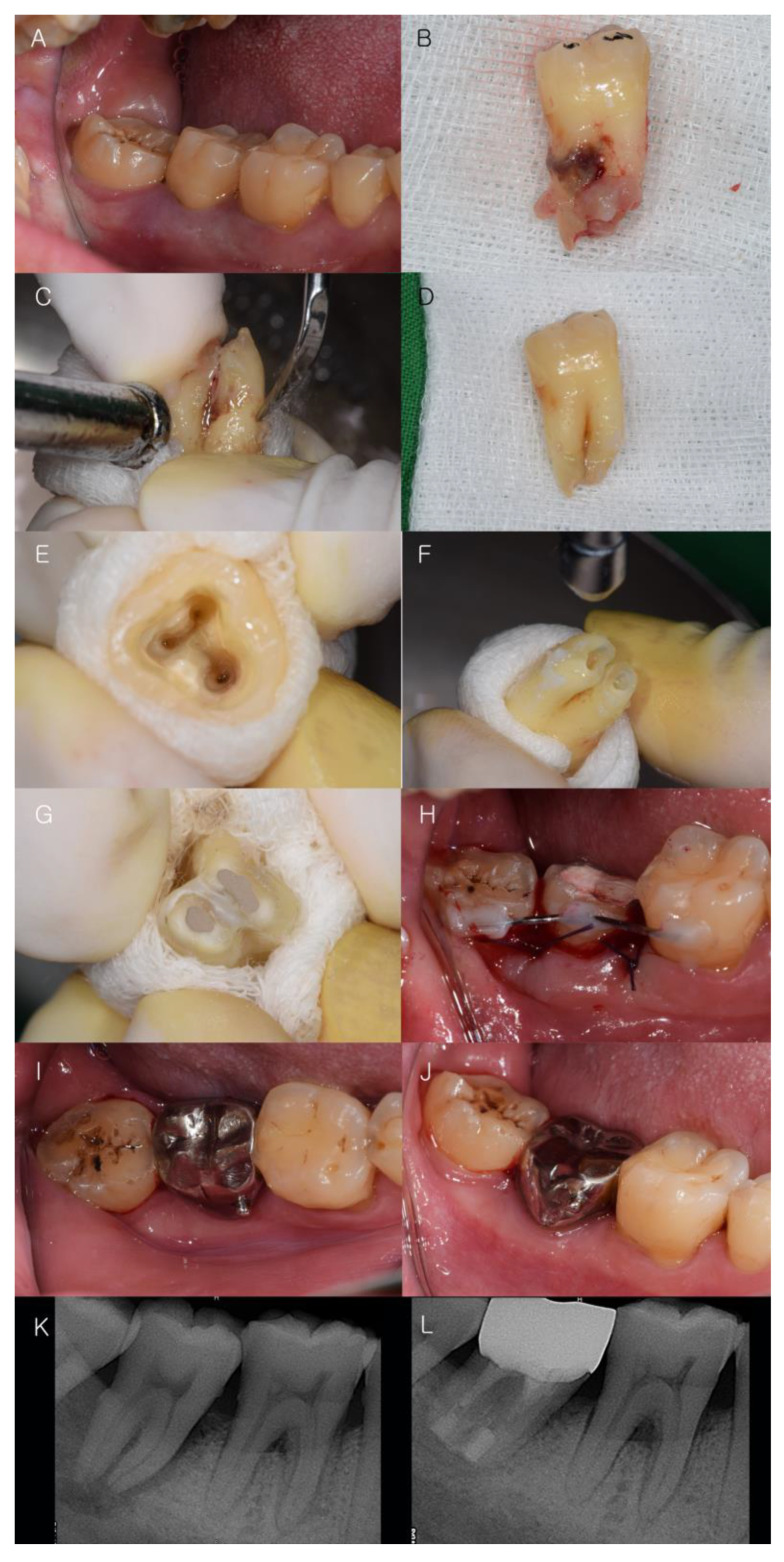
Surgical procedure of IR. (**A**–**D**) The tooth was extracted with atraumatic force. Granulation tissue and calculus on root surface were removed; (**E**–**G**) A 3-mm root-end preparation was achieved. Retrograde filling was performed with MTA and 4-META/MMA-TBB dentin-bonded resin; (**H**–**J**) The tooth was splinted with the adjacent teeth for 12 weeks. Prosthetic restoration was made after 3 months; (**K**,**L**) The radiographic images before and after IR. IR: intentional replantation, MTA: mineral trioxide aggregate, 4-META/MMA-TBB: The 4-methacryloxyethyl trimellitate anhydride/methyl methacrylate-tri-n-butylborane.

**Figure 4 medicina-59-00040-f004:**
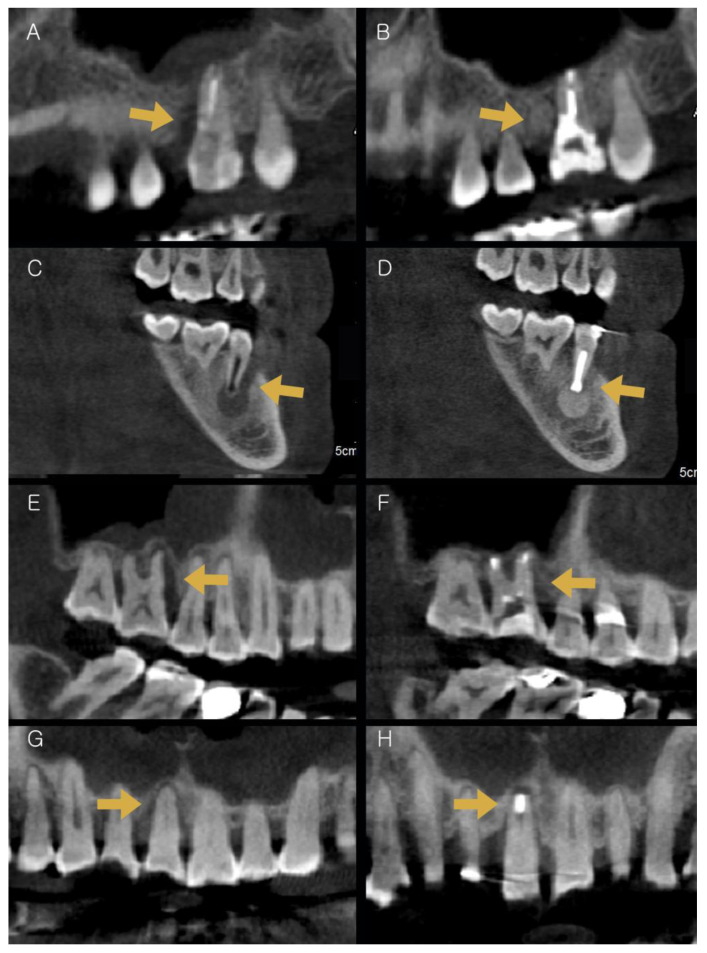
Images of CBCT between before and after IR. (**A**,**C**,**E**,**G**) CBCT images before IR; (**B**,**D**,**F**,**H**) CBCT images after IR (Arrow: bone healing area); IR: intentional replantation, CBCT: Cone–Beam Computed Tomography.

**Table 1 medicina-59-00040-t001:** General characteristics.

Characteristics	Patients (*N* (%))
*N*	14 (100.00)
Age (years) (Mean (SD))	46.79 (9.95)
Male	42.88 (3.90)
Female	52.00 (2.27)
Sex	
Male	8 (57.10)
Female	6 (42.90)
Jaw position	
Upper anterior	1 (7.10)
Upper posterior	4 (28.60)
Lower anterior	0 (0.0)
Lower posterior	9 (64.30)
Systemic disease	
Hypertension	1 (7.10)
Chronic rhinitis	1 (7.10)
Thyroid disease	1 (7.10)
None	11 (78.60)
Smoking	
Past	0 (0.0)
Not	14 (100.00)

**Table 2 medicina-59-00040-t002:** The methods during the IR procedure.

Methods	Patients (*N* (%))
Pre-operation orthodontic procedure	
Ni-Ti wire & button	2 (14.3)
Separating ring	4 (28.6)
None	8 (57.1)
GBR	
DFDBA	9 (64.29)
None	5 (35.71)
Retro-filling material	
MTA	11 (78.57)
4-META/MMA-TBB dentin-bonded resin	3 (21.43)

Ni-Ti: Nickel-Titanium, GBR: Guided bone regeneration, DFDBA: Demineralized freeze-dried bone allograft, MTA: Mineral trioxide aggregate, 4-META/MMA-TBB: The 4-methacryloxyethyl trimellitate anhydride/methyl methacrylate-tri-n-butylborane.

**Table 3 medicina-59-00040-t003:** Clinical changes between pre- and post-operation.

Variables	Pre-Operation	Post-Operation	*p*-Value
Pain	4.71 (2.84)	1.00 (1.18)	0.001 *
Mobility	1.36 (1.34)	0.29 (0.47)	0.010 *
PD	5.60 (2.02)	2.85 (1.21)	0.001 *
BOP	3.50 (1.56)	0.79 (0.70)	0.001 *

Values are presented as mean (standard deviation). Pain, Visual Analogue Scale (0–10); Mobility, Miller’s classification (1–3); BOP: bleeding on probing, Mean value on 6 sites showing bleeding; PD: Probing depth, Mean value of probing depth on 6 sites; * *p* < 0.05.

**Table 4 medicina-59-00040-t004:** Changes in bone height before and after IR.

Variables	Pre-Operation	Post-Operation	*p*-Value
Bone height (Mx)	43.29 (28.45)	47.29 (33.04)	0.109
Bone height (Mn)	115.50 (42.98)	113.55 (39.45)	0.551

Values are presented as mean (standard deviation). IR, Intentional replantation; Mx, Maxilla; Mn, Mandible.

## Data Availability

The data presented in this study are available on request from the corresponding author. The data are not publicly available due to privacy reasons.

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
