# Peer review of "Assessment of Bone Height Changes Based on the Cone–Beam Computed Tomography Following Intentional Replantation for Periodontally Compromised Teeth"

_medicina, 2022, doi:10.3390/medicina59010040_

Round 1
Reviewer 1 Report
Method section:
* Examination criteria used for PD ?
Pain was measured using a visual analog scale 91 (VAS) ?. Add more explanation / add reference .
Author Response
Thank you for your kind comment. We have tried to do our best to answer your request. Please refer to the attached file for details.
Yours faithfully,
Bongju Kim, PhD and Jung-Tae Lee, DDS, PhD

Reviewer 2 Report
The study seems interesting and genuine, however the following points should be addressed by authors to improve the overall quality of the manuscript:
- The abstract should include a short statement on the current gap in literature relative to this topic.
- The abstract included much of technical details, please try to shorten it as much as you can.
- Introduction section is too short, please add more relevant and updated literature information.
- The authors should add the null hypothesis/hypotheses at the end of the introduction section.
- Since measurements were done by 2 investigators, have authors done inter-examiner variability test? please elaborate in the main text.
- Please provide reasons for using the described statistical analysis test? were data normally distributed?
- Conclusion section can be expanded and summarized in bullets.
Author Response

(The authors gave the same response as above.)
